# Improved Stabilization and In Vitro Digestibility of Mulberry Anthocyanins by Double Emulsion with Pea Protein Isolate and Xanthan Gum

**DOI:** 10.3390/foods12010151

**Published:** 2022-12-28

**Authors:** Yan Cao, Chenxing Liu, Shengming Lu, Yoshiharu Fujii, Jiaxiu Jin, Qile Xia

**Affiliations:** 1State Key Laboratory for Managing Biotic and Chemical Threats to the Quality and Safety of Agro-Products, Zhejiang Provincial Key Laboratory of Fruit and Vegetables Postharvest and Processing Technology, Ministry of Agriculture and Rural Affairs Key Laboratory of Post-Harvest Handling of Fruits, Institute of Food Science, Zhejiang Academy of Agricultural Sciences, Hangzhou 310021, China; 2United Graduate School of Agriculture, Tokyo University of Agriculture and Technology, Tokyo 183-8509, Japan

**Keywords:** double emulsions, mulberry anthocyanins, pea protein isolate, xanthan gum, stabilization, in vitro digestibility

## Abstract

There is significant evidence that double emulsion has great potential for successfully encapsulating anthocyanins. However, few research studies are currently using a protein-polysaccharide mixture as a stable emulsifier for double emulsion. This study aimed to improve the stability and in vitro digestibility of mulberry anthocyanins (MAs) by employing a double emulsion composed of pea protein isolate (PPI) and xanthan gum (XG). The influence of various XG concentrations (0%, 0.2%, 0.4%, 0.6%, 0.8%, 1.0%) and different temperatures (5 °C, 25 °C, 45 °C, 65 °C) on the physical stability and the thermal degradation of MAs from double emulsions were investigated. In addition, the physicochemical properties of double emulsions and the release performance of MAs during in vitro simulated digestion were evaluated. It was determined that the double emulsion possessed the most stable physical characteristics with the 1% XG addition. The PPI-1% XG double emulsion, when compared to the PPI-only double emulsion, expressed higher thermal stability with a retention rate of 83.19 ± 0.67% and a half-life of 78.07 ± 4.72 days. Furthermore, the results of in vitro simulated digestion demonstrated that the MAs in the PPI-1% XG double emulsion were well-protected at oral and gastric with ample release found in the intestine, which was dissimilar to findings for the PPI-only double emulsion. Ultimately, it was concluded that the double emulsion constructed by the protein-polysaccharide system is a quality alternative for improving stability and absorption with applicability to a variety of food and beverage systems.

## 1. Introduction

Mulberry is an edible and medicinal homologous fruit that provides anti-thrombosis, anti-inflammatory and neuroprotection [1]. Mulberry anthocyanins (MAs) contain the functional components in mulberry, mainly in the forms of Cyanidin 3-O-Glucoside and Cyanidin-3-O-Rutinoside [2]. Anthocyanins are polyphenolic compounds of the flavonoid family and are polyhydroxy derivatives of 2-phenylchromen (flavonoids). Additionally, the presence of the active phenolic hydroxyl groups contributes to the instability of anthocyanins [3]. Previous research suggests that MAs are susceptible to the loss of food processing abilities due to temperature, oxygen, metal ions, enzymes, and other factors [4]. Pharmacokinetic findings indicated that MAs can be absorbed through the stomach and intestine. However, their inability to return from the chemically bound state to the flavonoid cation via methylation, conjugation, glucose sulfation and other metabolic forms also affects digestive absorption [5].

In recent years, researchers have been considering double emulsion as an effective method for protecting anthocyanins. Double emulsion refers to the formation of a water-in-oil (W_1_/O) emulsion. The double emulsion is complete after the inner water phase (W_1_) is dispersed in the oil phase (O), and then the water-in-oil (W_1_/O) emulsion is dispersed in the outer water phase (W_2_). Therefore, a two-sided, three-phase stable structure is formed [6]. The unique structure of the double emulsion enables the effective encapsulation of water-soluble substances such as anthocyanins [7], ensuring the efficient delivery of active substances during digestion [8]. However, the emulsifier of conventional double emulsion is a surfactant. Consequently, this traditional method exhibits creaming, flocculation, and physical instability associated with internal waters [9]. Furthermore, to enhance stability, more practical emulsification is required. The utilization of a protein-polysaccharide mixture as the natural emulsifier in the external water phase through complementary hydrophilic and hydrophobic approaches could obtain the stable and two well-wetting double emulsion [10]. Similarly, Shaddel et al. presented a new idea to improve the stability of water-soluble active substances by preparing W_1_/O/W_2_ double emulsions which efficiently encapsulated anthocyanins using a gelatin-arabic gum mixture [11].

Utilizing this interaction between proteins and polysaccharides is a common method for stabilizing to stabilize emulsion-embedded systems. The stability of emulsions can be vastly enhanced by an appropriate protein-to-polysaccharide ratio, temperature and pH conditions [12]. When the pH of the system is close to the isoelectric point (PI) of protein, the electrostatic repulsion of the protein adsorption layers decreases, which allows more compact packing and stronger attractive interfacial interactions so that coalescence and flocculation occur [13]. Additionally, proteins and polysaccharides are able to stabilize the emulsion interface without interaction. For example, while proteins are adsorbed at the oil-water interface, polysaccharides increase the viscosity of the aqueous phase and provide stabilization [14]. Pea Protein Isolate (PPI) is of significant interest due to its low cost, high nutritional value, and low allergenicity. Furthermore, PPI is known to have desirable solubility, foamability, emulsification and gelation properties under suitable conditions, and has the potential for the development of functional foods [15]. However, Liang et al. found that PPI’s emulsification ability and emulsion stability were very poor near the isoelectric point and further requires the viscosity of polysaccharides to improve its stability [16]. Kim et al. [17] reported that sodium alginate improved the emulsion stability of pea protein isolate at pH 6.6 in that creaming precipitation after 21 days of storage was not present. Contrarily, xanthan gum (XG) has substantial potential for improving the stability of proteins. XG is an extracellular anionic polysaccharide derived from microbial decomposition with the high viscosity. Studies have found that XG is relatively stable under extreme conditions, including acid and alkali, therefore suggesting the ability to reduce the immiscibility between oil and water phase and improve the emulsification and stability of food system [18]. Bouyer et al. [14] reported that emulsions composed of XG and β-lactoglobulin possess dominant stabilization compared to arabic gum and β-lactoglobulin compositions for emulsions. Therefore, it can be speculated that the emulsion composed of XG and protein can also be used to encapsulate and deliver other active substances.

There are no current reports about using PPI and XG compositions for double emulsions. The utilization of the emulsification presented by PPI and the viscosity from XG, as well as the interaction between them, provides favorable prospects for improving the emulsion stability and encapsulation of water-soluble active substances. This study aimed to examine the influence of XG addition and temperature on the stability of double emulsion and the thermal degradation of MAs. Besides, in order to understand the delivery of the MAs embedding system, the in vitro digestibility of PPI-XG stabilized emulsion was also evaluated.

## 2. Materials and Methods

### 2.1. Materials

Mulberry Anthocyanins (Mas) powder (25% anthocyanins), xanthan gum (XG) and Polyglycerol polyricinoleate (PGPR) were purchased from Duolong Biotechnology Co., Ltd. (Nanjing, China). Peanut oil was bought from a local market. Pea protein isolate (PPI), α-Amylase from porcine pancreas, pepsin from porcine stomach mucose, trypsin from porcine pancreas, lipase from porcine pancreas, bile salt from pig, dipotassium hydrogen phosphate (K_2_HPO_4_), and 1,1-diphenyl-2-picrylhydrazyl (DPPH) were attained from YuanYe Biotechnology Co., Ltd. (Shanghai, China). Additionally, 2,4,6-Tris(2-pyridyl)-1,3,5-triazine (TPTZ), Nile blue, Nile red, 6-hydroxy-2,3,7,8-tetramethylchroman-2-carboxylic acid (Trolox), ferric chloride, and analytical grade of methanol were acquired from Aladdin Biochemical Technology Co., Ltd. (Shanghai, China). Other analytical grade of chemicals were provided by the China National Pharmaceutical Group Co., Ltd. (Beijing, China).

### 2.2. Preparation of PPI-XG Hydrocolloids

PPI-XG hydrocolloids were used as the hydrophilic emulsifiers of the outer water phase during the production of the double emulsions. In brief, the PPI solution had a final concentration of 2% (*w*/*v*) while the XG solution had the final concentrations of 0%, 0.2%, 0.4%, 0.6%, 0.8%, and 1% (*w*/*v*) respectively. Finally, PPI-XG hydrocolloids with a volume ratio of 1:1 were obtained.

### 2.3. Production of the Double Emulsion Encapsulated MAs

The MAs-encapsulated double emulsion was described as W_1_/O/W_2_ and was constructed according to the methods specified by Shaddel et al. [11]. Furthermore, W_1_ was the inner water phase composed of MAs aqueous solution, O was the oil (peanut oil) phase, and W_2_ was the outer water phase composed of PPI-XG mixtures. The primary W_1_/O emulsion was prepared with addition of a 30 g MAs aqueous solution to a 70 g oil phase (contained 8 g PGPR) using a high-speed blender (FJ 200-SH, Specimen Model Factory, Shanghai, China) for 3 min at 18,000 rpm. Afterward, the W_1_/O emulsion (40 g) was dispersed into the outer-phase solution (60 g) and the mixture was homogenized for 1.5 min at 10,000 rpm to obtain a coarse emulsion.

### 2.4. Encapsulation Efficiency and Creaming Stability of Double Emulsion

In order to determine the encapsulation efficiency (EE%), the content of the total MAs was measured using the pH differential method. The encapsulation efficiency of MAs in double emulsions was evaluated by determining the initial content of total MAs and the content of free MAs unloaded using Equation (1) by the method expressed by Ge et al. [19] with few modifications.
(1)$$\mathrm{EE}(\%)=\frac{\mathrm{Total}\;\mathrm{MAs}-\mathrm{Free}\;\mathrm{MAs}}{\mathrm{Total}\;\mathrm{MAs}}\times 100$$

The relative stability of the emulsion was assessed by examining the creaming index during the storage period [20]. Emulsions were placed into 15 mL glass tubes and sealed to avoid evaporation and pollution. The sample tubes were stored at various temperatures of 5 °C, 25 °C, 45 °C, 65 °C for 20 days. The creaming stability could be determined through using the creaming index which utilized the total height of the emulsion (HE) and the height of the serum layer (HS) using Equation (2) as follows:(2)$$\mathrm{Creaming}\;\mathrm{index}\;(\%)=\frac{\mathrm{Serum}\;\mathrm{layer}\;\mathrm{height}}{\mathrm{Total}\;\mathrm{height}\;\mathrm{of}\;\mathrm{emulsion}}\times 100$$

### 2.5. Particle Size, Zeta Potential and Microstructure

The mean particle size and the zeta potential of double emulsions were analyzed with Zetasizer Nano instrument (ZEN 3500, Malvern Instrument Ltd., Worcestershire, UK) and dynamic light scattering (DLS) technology. Previous research suggests that these parameters have relevance to emulsion characteristics including appearance, viscosity, and time stability [21]. Using an optical microscope (Olympus CX31, Olympus Imaging Corporation, Tokyo, Japan), with a VT-sanp 300 digital camera system (Vision Datum Technology CO., Ltd., Hangzhou, China), the microstructure of double emulsions was observed. The sample emulsion was dripped on a clean glass slide with a coverslip and observed at 40× objective magnification. Sample emulsions were diluted 100 times and then taken to the quartz cuvette cell with a square aperture. The refractive index of oil and water phase was set to 1.456 and 1.33 respectively, the absorption index was set to 0.01 [22].

Fluorescent imaging of the double emulsions during the in vitro digestion period was obtained by employing confocal laser scanning microscopy (CLSM) with the methods described by Ma et al. [22]. As the fluorescent dyes, Nile red (0.1% in methanol, *w*/*v*) and Nile blue (0.1% in methanol, *w*/*v*) were added to the emulsion samples. The micro-morphology of the emulsion droplets was observed under 40× objective magnification.

### 2.6. In Vitro Simulated Digestion of Double Emulsions Encapsulated MAs

Double emulsions were used to evaluate the properties of in vitro simulated digestion. The in vitro digestion model simulated digestion of the mouth, stomach, and intestine according to the methods detailed by Minekus et al. [23] and the United States Pharmacopoeia (USP) with a few modifications.

Simulated mouth stage (0–60 min): The simulated saliva fluid (SSF) was obtained by dissolving α-Amylase in a calcium chloride solution to obtain enzymatic activity at 100 U/mL and the pH of the SSF was adjusted to 7.0. The SSF was added to the initial double emulsion and incubated at 37 °C for 60 min.

Simulated gastric stage (60–180 min): The simulated gastric fluid (SGF) main component was pepsin. The pH of the SGF was adjusted to 2.5. The resulted mixture of the SSF (10 mL) was mixed with SGF (10 mL) and incubated at 37 °C for 2 h.

Simulated intestinal stage (180–360 min): The simulated intestinal fluid (SIF) contained trypsin, lipase and bile salt. The SIF was added to the resulted mixture from simulated gastric phase (10 mL) and incubated at 37 °C for 3 h.

### 2.7. Thermal Degradation of Mas from Double Emulsions at Different Temperatures

The thermal stability of the double emulsions during storage were evaluated by the retention rate of MAs and the first-order degradation kinetic model. The double emulsions were stored at different temperatures of 4 °C, 25 °C, 45 °C, 65 °C for 20 days. The MAs retention rate was determined using Equation (3). Both the reaction rate (k) and half-life (t_1/2_) from the MAs degradation kinetic model were calculated using Equations (4) and (5) according to previous research by De et al. [24].
(3)$$\mathrm{MAs}\;\mathrm{retention}\;(\%)=\frac{C_{t}}{C_{0}}\times 100$$

C_t_ is the concentration of MAs after storage time t (day) and C_0_ is the initial MAs concentration:(4)$$\ln\frac{C_{t}}{C_{0}}=-\mathrm{kt}$$
(5)$$t_{1/2}=\frac{\ln2}{k}$$
k is the reaction rate related to the temperature and t is storage time.

### 2.8. Release of MAs in Double Emulsions during In Vitro Simulated Digestion

The release rate of MAs (R%) in the double emulsions during in vitro digestion period were calculated with Equation (6) according to the work of Xu et al. [25]. The encapsulation efficiency (EE%) was determined using Equation (1).
R% = 100 − EE%(6)

### 2.9. Oxidative Stability of Double Emulsions during In Vitro Simulated Digestion

Oxidative stability of the double emulsions encapsulated MAs during in vitro digestion period was evaluated by determining the DPPH free radical scavenging activity (DPPH-RSA) and ferric ion reducing antioxidant power (FRAP).

DPPH-RSA was determined according to the method described by choi et al. [26] with a few modifications. The 0.1 mL diluted sample emulsion was mixed with 2.9 mL of 100 mol/L DPPH methanol solution. The absorption of the sample solution was measured at 517 nm by a microplate reader (Spark 20 M, Tecan Austria GmbH, Grodig, Austria) and the DPPH-RSA was expressed as equivalents of Trolox (μmol TE/g) per g of sample emulsion.

FRAP was determined according to the methods suggested by Benzie et al. [27] with a few modifications. The 0.3 mL diluted sample emulsion was then added to 3 mL FRAP working solution. The absorption of the mixed solution was measured at 593 nm using a microplate reader (Spark 20 M, Tecan Austria GmbH, Grodig, Austria) and the FRAP was expressed as equivalents of Trolox (μmol TE/g) per g of sample emulsion.

### 2.10. Statistical Analysis

All the measurements were conducted in triplicate for each sample, and the results were expressed as the mean (M) ± standard deviation (SD). Statistical testing was performed using SPSS statistical software (SPSS, 21.0). Analysis of variance (ANOVA) and Tukey’s HSD test were adopted to determine the significant level (*p* < 0.05) among the mean values.

## 3. Results

### 3.1. Effects of Various XG Concentrations on the Physical Properties of PPI-XG Double Emulsions Encapsulated MAs

#### 3.1.1. Particle Size and Microstructure

The size of the droplets in an emulsion represented the physical stability of the emulsion. A smaller particle size suggested improved stability [9]. The physical properties and optical microstructures of the double emulsions composed of PPI (2%, *w*/*v*) and different concentrations of XG (0%, 0.2%, 0.4%, 0.6%, 0.8% and 1%) were showed in Table 1 and Figure 1A,B. Interestingly, when the concentration of XG was changed from 0% to 0.2%, the particle size of the emulsion increased from 9.20 ± 0.21 μm to 9.83 ± 0.17 μm (Table 1). Besides, the optical microscopy images (Figure 1B) exhibited large droplets in the emulsions with obvious aggregation. A likely reason for these results is that the low concentration of XG is unable to span the surface of the protein-coated emulsion droplets completely, resulting in reduced the stability of the emulsion through protein molecular bridging flocculation or aggregation between oil droplets [28]. Furthermore, as the concentration of XG increased from 0.4% to 1%, the particle size of the emulsion decreased from 7.40 ± 0.23 μm to 3.80 ± 0.33 μm (Table 1), Moreover, the optical microscopy images (Figure 1B) showed that the emulsion changed from large aggregated droplets to the smaller homogeneous droplets, which is consistent with the results of Abdolmaleki et al. [29]. Results were potentially caused by the enhancement of XG concentration with increased viscosity. Additionally, a larger interfacial thickness was formed. Therefore, a protective layer formed around the droplets, reducing the likelihood of inter-droplet motion and collision, resulting in a decreased emulsion particle size [30].

#### 3.1.2. Zeta Potential

Zeta potential reflects the charged aspect of the emulsion droplet. Yi et al. [31] reported a positive correlation between zeta-potential and emulsion stability, further suggesting that a higher absolute zeta potential will produce a stronger emulsion capable of resisting aggregation and flocculation during storage. Additionally, previous studies noted that an emulsion composed of 2% (*w*/*v*) whey protein isolate and 0.5% (*w*/*v*) xanthan gum were stable for 15 days and had a zeta-potential at −34.90 ± 1.31 mv [32]. As shown in Table 1, the variation in the zeta potential of the double emulsions ranged from −25.23 ± 0.21 mV to −32.58 ± 0.31 mV, respectively, with the XG concentration change from 0% to 1%. The pH of all samples was found to be 6, which is significantly higher than the isoelectric point (PI) of PPI (PI of 4.6). With respect to the increase in XG concentration, the absolute value of the potential increased gradually. It indicated that when the pH of the emulsion exceeded the PI of the protein, the protein and the anionic polysaccharide in the emulsion system were negatively charged, and the addition of the anionic polysaccharide correlated with an increase in the net charge number of the emulsion. Furthermore, the electrostatic repulsion caused by the increased charge could effectively prevent the prevent the collision and aggregation of droplets while improving the emulsion’s stability [33].

#### 3.1.3. Encapsulation Efficiency

The double emulsion produced by using the protein-polysaccharide has potential for effectively encapsulating anthocyanin. Kanha [34] successfully produced double emulsions using anthocyanins and reported encapsulation efficiencies of 84.9% and 94.7% by using 4% gelatin-4% arabic gum, and 0.5% chitosan-0.1% Carboxymethyl cellulose. In this study, the encapsulation efficiencies of the PPI-XG double emulsions containing different XG concentrations (0%, 0.2%, 0.4%, 0.6%, 0.8% and 1%) were 44.57 ± 0.93%, 51.5 ± 0.87%, 62.66 ± 0.79%, 68.52 ± 0.81%, 83.02 ± 0.70%, and 88.52 ± 2.05%, respectively, reported in Table 1. It was determined that the concentration of XG is positively correlated with the encapsulation efficiency of MAs, which was consistent with the results based on particle size and zeta potential. The double emulsion with a 1% XG concentration was found to have the highest encapsulation efficiency of 88.52 ± 2.05% for MAs. Similarly, Sun [35] reported a significant increase at the encapsulation efficiency as XG concentration exceeded 0.2%. According to the previous study, XG is a molecule with high molecular weight, charge density and rigidity, which can be adsorbed on the surface of protein droplets, resulting in increased spatial repulsion between droplets [36].

#### 3.1.4. Creaming Index

The apparent morphology and creaming index of the double emulsions composed of various XG concentrations and 2% PPI were shown in Figure 1A and Figure 1B, respectively. Creaming index reflects the stability of emulsion, with higher values indicating instability. The creaming phenomenon did not appear in the double emulsions containing 0.8% and 1% XG. Furthermore, the emulsions containing 0%, 0.2%, 0.4% and 0.6% XG displayed creaming indexs of 74.33 ± 1.53%, 45.00 ± 1.00%, 31.67 ± 1.53% and 15.33 ± 1.15%, respectively. The results suggested that the double emulsions without XG had the most severe creaming. However, the creaming index gradually decreased to 0 when the concentration of XG increased. Previous studies have shown that a higher concentration of XG could improve the electrostatic repulsion with protein and effectively prevent emulsion flocculation caused by the collision of droplets. Xu et al. found that when 0.4% XG was added, the electrostatic repulsion of the emulsion composed by rice protein and XG was significantly increased [37]. According to the research of Soumia et al. [38], no precipitation was observed in the emulsion with addition of a 0.75% XG and sodium caseinate solution, suggesting that XG improved the interfacial stability of the emulsion.

### 3.2. Effect of Different Temperatures on the Storage Stability of Double Emulsions

#### 3.2.1. Emulsion Morphology and Creaming Index

The changes in the creaming index of double emulsions composed of PPI-only and PPI-1% XG at 5 °C, 25 °C, 45 °C and 65 °C from 0 to 20 days and the apparent morphology after 20 days were shown in Figure 2A,B. After 20 days, significant creaming and delamination occurred in the PPI-only double emulsions at all temperatures (Figure 2A). However, no flocculation and delamination were observed at 5 °C and 25 °C for the PPI-1% XG double emulsions, indicating the stability of emulsions. In contrast, obvious flocculation and delamination from the water-oil interface was observed at 45 °C and 65 °C. The creaming index of PPI-only double emulsion at various temperatures, 5 °C, 25 °C, 45 °C, and 65 °C, increased by 20.33%, 27.89%, 33.66%, and 33.00%, respectively. Compared with the fresh emulsion, the increase in temperature significantly affected the interfacial stability of the PPI-only double emulsions. The creaming index of PPI-1% XG double emulsions at different temperatures increased compared to those of the original emulsions: 0%, 0%, 5.33% and 18.66%, respectively. The results indicated emulsions with the 1% XG addition were more stable at 5 °C and 25 °C, while the 45 °C and 65 °C conditions had reduced stability. McClement et al. [9] found that the thickening of the whey layer in emulsions is due to high temperatures and the prolongation of storage time causing phase separation. It may be that as the larger particles in the emulsion are absorbed into the oil layer, the smaller particles are dispersed into the whey layer. According to the previous research, it was found that whey protein-XG mixture could be jointly adsorbed onto the surface of oil droplets by the addition of appropriate concentrations of XG. It was further suggested that this addition increased the film thickness at the oil-water interface and the viscosity of the aqueous phase significantly, thus inhibiting the thermodynamic motion of the droplets [39].

#### 3.2.2. Microstructure, Particle Size and Zeta Potential

The microstructure, particle size, and zeta potential of the double emulsions composed of PPI-only and PPI-1% XG at 5 °C, 25 °C, 45 °C and 65 °C after 20 days were shown in Figure 3A,B. As temperature increased, the particle size of the PPI-only double emulsions increased significantly compared to the PPI-1% XG double emulsions. Additionally, the emulsions gradually showed the aggregation of large irregular droplets at 45 °C and 65 °C. Considering that the emulsion is a thermodynamically unstable, the rise of temperature likely promotes the Brownian motion and the collision of particles in emulsion, which causes the aggregation and rearrangement of PPI molecules at the oil-water interface, thus accelerating the instability of emulsion [40]. At 5 °C and 25 °C, the particle size of PPI-1% XG double emulsion did not increase significantly, which indicated that the viscosity of XG did not change significantly within a certain temperature range. In relation, the gel network formed by the protein restricts the accumulation of oil droplets [41]. At 45 °C and 65 °C, the droplets of PPI-1% XG double emulsions increased to 5.62 ± 0.21 μm and 6.30 ± 0.13 μm, respectively, further suggesting that the high temperature damaged the network structure of the protein-polysaccharide mixture and caused aggregation of the droplets.

Meanwhile, zeta potentials of the two double emulsions showed similar results. The absolute value of the zeta potential in PPI-only double emulsions showed a gradual decrease when the temperature was increased from 5 °C to 65 °C. When the temperature reached 65 °C, the PPI-only double emulsion expressed the lowest absolute value of zeta potential (19.7 mV, less than 20 mV), indicating that high temperature potentially damaged the structural and charge properties of PPI and caused the reduced stability of the emulsion. At temperatures of 5 °C and 25 °C, the absolute value of zeta potential in the PPI-1% XG double emulsions remained at 31.74 ± 0.32 mV and 32.03 ± 0.28 mV. As the temperature increased from 45 °C to 65 °C, the absolute value of zeta potential in the PPI-1% XG double emulsions decreased from 29.15 ± 0.30 mV to 28.00 ± 0.44 mV (less than 30 mV), suggesting that the high temperature had an inhibitory effect on the stability of the PPI-XG double emulsion system. Additionally, the high viscosity of XG did not prevent the abating of viscous resistance caused by the strong thermal motion between molecules and resulted in a destabilization of the emulsions [42].

### 3.3. Thermal Degradation of MAs at Different Temperatures

The thermal stability of MAs in PPI and PPI-1% XG double emulsions stored at different temperatures of 5 °C, 25 °C, 45 °C, and 65 °C for 20 days was evaluated by the retention of Mas, as well as degradation kinetics-related parameters (degradation rate constant k, Pearson’s correlation coefficient R^2^ and half-life t_1/2_), as shown in Figure 4 and Table 2. The retention rate of MAs in both emulsions decreased with increasing temperatures during 20 days. For the PPI-only double emulsion, it possessed the highest retention rate of 61.66 ± 1.47% for MAs was found at 5 °C and the lowest retention rate of 40.32 ± 1.28% for MAs was found at 65 °C after 20 days. For the PPI-1% XG double emulsion, less MAs were degraded at 5 °C with a retention rate of 83.19 ± 0.67% after 20 days. Relatively, the retention rate of MAs in the PPI-1% XG double emulsion was lower at 45 °C and 65 °C after 20 days. Previous studies reported that anthocyanin from acai berries degraded by 38% after 10 days in a nano double emulsion, but many large droplets were observed in addition to the collapse of the internal water phase [43]. In this step, effective encapsulation of MAs was achieved and a high retention rate during storage was maintained with the PPI-1% XG stabilized double emulsions.

According to the primary degradation kinetics, the degradation of MAs in both emulsions showed high Pearson correlation coefficients (r^2^ > 0.96), which is consistent with the previous findings [24]. Furthermore, as the storage temperature increased from 5 °C to 65 °C, the degradation rate K of MAs in both emulsions increased while the half-life (t_1/2_) reduced, exhibiting that the high temperature promoted the degradation of anthocyanins, which was similar to the findings reported by Nayak et al. [44]. The results in Table 2 showed that the K values of MAs in PPI-1% XG double emulsion was smaller than those found in the PPI-only double emulsion at the same storage temperature. Additionally, the half-life of MAs degradation was prolonged in the PPI-1% XG double emulsion compared to the PPI-only double emulsion. The half-lives of MAs in the PPI-1% XG double emulsions positively correlated with the temperature increase: 47.75 days, 25.04 days, 17.48 days, and 13.90 days respectively compared with those in the PPI-only double emulsions. It was determined that high temperatures promoted the unfolding of the protein structure, while also increasing mutual attraction of the more reactive groups (such as nonpolar or sulfhydryl groups) [45]. Moreover, the temperature increase promoted the thermal movement of the emulsion as well as the aggregation of droplets, ultimately resulted in an unstable system. It is understood that the high viscosity of XG could slow down the thermal motion and mutual collision of droplets in emulsions. However, the continuous warming could lead to the rearrangement of the ordered structure of XG, resulting in significant flocculation and aggregation of the droplets [46].

### 3.4. In Vitro Digestive Properties of Double Emulsions

#### 3.4.1. Particle Size and Micromorphology

The changes in the microstructure and mean particle size in the PPI-only and PPI-1% XG double emulsions encapsulated MAs at each stage of in vitro digestion were exhibited in Figure 5A and Figure 5B, respectively. The mean particle size of the PPI-only double emulsion increased significantly after simulated oral digestion, while the mean particle size and microstructure of the PPI-1% XG double emulsion did not change significantly. Vingerhoeds et al. reported that the mucin and mineral ions in the SSF could not resist the electrostatic effect of emulsion particles through repulsion and bridging flocculation when only whey protein or at a low concentration of XG [47].

The average particle size of the double emulsions composed of PPI-only and PPI-1% XG increased after simulated gastric digestion (SGG), as shown in Figure 5B. Confocal laser micrographs exhibited that the PPI-only double emulsion posed irregularly large droplets following fusion with significant aggregation between the emulsion droplets (Figure 5A). Similarly, the double emulsion of grape skin anthocyanin encapsulated by soybean protein isolate underwent significant flocculation in SGG [26], which might be caused by physicochemical changes (such as diffusion, heat transfer, solubilization in gastric fluid) and hydrolysis (such as pepsin that adversely affected the stability of proteins and polysaccharides) [48]. However, the PPI-1% XG double emulsion did not exhibit large aggregation of emulsion droplets due to the significant increase of XG concentration. Potentially, these results are because the addition of XG reduced the flocculation of droplets and the MAs in the inner layer of the emulsion W_1_/O were protected.

The average particle size of both double emulsions decreased after simulated intestine digestion (SIG), as displayed in Figure 5B. The average particle size of PPI-1% XG double emulsion decreased to a minimum of 2.72 ± 0.31 μm, and confocal laser micrographs showed digestion into smaller hollow oil droplets (Figure 5A). These results suggest the full release of MAs from the inner W_1_/O layer of the emulsion. During the SIF digestion, lipids hydrolyzed to produce different types of particles, including undigested fat droplets and mixed micelles (free fatty acids, diglycerides, and phospholipids). All of these particles disrupted the internal and external structure of the emulsion, while the high viscosity of XG slowed the collision of oil droplets with fat micelles and lead to the smaller particle size [49].

#### 3.4.2. Zeta Potential

Zeta potential is used to explain the effect of simulated digestion on the charge characteristics of the double emulsions. After SSF digestion, zeta potential of both the PPI-only and PPI-1% XG double emulsions were negative and decreasing. It is suggested that less salt ions in oral saliva caused a reduction in the zeta potential of droplets in the emulsion system through electrostatic shielding or ion binding. Additionally, changes in the charge are thought to be a result of mucins absorption on the droplet surface [50].

After SGF digestion, the charge of the PPI-only and PPI-1% XG double emulsions changed from negative to positive at 7.60 ± 0.30 mV and 1.86 ± 0.08 mV, respectively. This phenomenon could be involved with the low pH (2.5) in SGF, reducing the negative charge on the droplet surface through the charge shielding caused by high ionic strength [51]. Previous studies have found that with the pH of SGF was less than the isoelectric point of PPI, positively charged proteins and their lipid droplet interfaces gravitated electrostatically towards the anionic polysaccharide molecules [52], thus reducing the negative charge of XG.

The initial positive charged shifted to a negative charge following SIF digestion. The shift in positive and negative charges and the decrease in zeta potential is thought to be due to the elevated pH in SIF. Furthermore, the anionic cholate induced a higher negative charge with PPI and XG. Moreover, droplets on the surface of the emulsion were absorbed by micellar components (such as digestive products of free fatty acids, bile salts or bioenzymes) in SIF, causing the change in charge strength [53]. In addition, the PPI-1% XG double emulsion with a lower negative charge of −20.46 ± 0.31 mV in SIF indicated that the presence of XG on the droplet surface increased the negative charge of emulsion [32].

#### 3.4.3. Release Properties of MAs for In Vitro Simulated Digestion

The release properties of MAs in the PPI-only and PPI-1% XG double emulsions were found to be different during each phase of in vitro simulated digestion (Figure 6). After SSF digestion, only 12.43 ± 2.70% of MAs were released from the PPI-only double emulsion, while the release rate of MAs from the PPI-1% XG emulsion was twice times as the PPI-only double emulsion, indicating the higher concentration of XG and PPI formed a viscous mesh structure protecting the inner layer of the emulsion from oral saliva erosion [54].

After SGF digestion, the release rate of MAs in the PPI-only emulsion and PPI-1% XG emulsion increased to 47.80 ± 2.10% and 27.55 ± 2.30%, respectively. The results indicated that the PPI-1% XG double emulsion presented a protective effect on the active ingredients at this stage. Potentially, the outer layer of the PPI-1% XG emulsion was covered by XG with dominant pH stability [55], thus resisting the acidic environment in SGF and reducing leakage fo MAs from the inner water phase.

The release rates of MAs from PPI-only and PPI-1% XG double emulsions increased to 60.13 ± 1.30% and 71.15 ± 3.10%, respectively, after digestion in SIF. We concluded the results from the in vitro release and confocal microscopy images that the double emulsion structures were severely broken by digestive enzymes, cholate, and alkaline conditions, which promoted the breakdown of the emulsion droplets and the release of anthocyanins from the internal water phase.

#### 3.4.4. Evaluation of Oxidative Stability of Double Emulsions

DPPH-RSA values and FRAP values were adopted to evaluate the oxidative stability of the double emulsions at each stage of simulated digestion (Figure 7). At the end of SSF digestion, the DPPH-RSA values and FRAP values in two emulsions showed minor changes compared with those of the original emulsions encapsulated with MAs. This change implies that while the amylase and protein in oral saliva promoted the aggregation of the emulsion droplets, there was no destruction of the outer structure of the emulsions, hence allowing the antioxidant properties of the MAs encapsulated in the interior to be protected.

At the end of SGF, the DPPH-RSA and FRAP values of the PPI-only double emulsion increased by 28.11% and 32.55%, respectively. Compared to the end of oral digestion, the increase in DPPH-RSA and FRAP values by the PPI-1% XG double emulsion were less than 15.00%. Based on the repot of Beermann et al., the significant increase in antioxidant ability of the protein-only double emulsion verified the exocytosis of MAs in the inner layer of the emulsion, which was associated with the hydrolysis of the protein in simulated gastric fluid [56]. Since anthocyanins were more stable under acidic conditions [57], the high antioxidant activity of free MAs in SGF were discharged.

However, the antioxidant capacity of the two double emulsions were reduced under SIF and the DPPH-RSA as well as FRAP values of the PPI-only double emulsion decreased significantly by 27.07% and 31.35%. According to the study of Hu et al., the structure of the PPI-only double emulsion probably has been disrupted seriously and free MAs released from the SGF and SIF were easier to degrade in the alkaline condition of intestine fluid [58].

## 4. Conclusions

This study used PPI and XG to stabilize double emulsions encapsulated MAs. The addition of XG significantly improved the stability of the emulsions compared to the PPI-only emulsion. The double emulsion with 1% XG addition showed the most stable physical properties, presenting the smallest particle size and creaming index as well as the highest absolute value of zeta potential and encapsulation efficiency. The PPI-only double emulsion was more sensitive to temperature, and the stability of the two emulsions decreased with increasing temperature. The PPI-1% XG double emulsion was relatively more stable at low-temperature storage of 5 °C and 25 °C. The retention rate and half-life of MAs in PPI-1% XG double emulsions improved at all temperatures compared to the PPI-only double emulsions, thus indicating that the PPI-1% XG double emulsions could improve the thermal stability of MAs. In the in vitro digestion model, the PPI-only double emulsion demonstrated a significant decrease in particle size, and absolute value of zeta potential. Interestingly, the PPI-1% XG emulsion presented smaller particle size, lower release rate of MAs, and antioxidant properties at the mouth and gastric. The addition of XG improved the stability of PPI and MAs’ biological activity. Both emulsions improved the release rate of MAs and reduced antioxidant properties at the intestine stage, suggesting that the alkaline environment of SIF and components such as enzymes, cholate promoted the collapse of the double emulsion structure and encouraged the free MAs. The novel observations have been found that the PPI-1% XG double emulsion improved MAs’ stability and in vitro digestive properties, which can provide a new idea for the construction of anthocyanins delivery system. Besides, studies on the stability of double emulsions at the different pH value and ionic strength will be conducted. In the future, the double emulsions composed of PPI and XG can be added to functional foods (such as mayonnaise, jam, and salad dressing) to improve their nutritional value.

## Figures and Tables

**Figure 1 foods-12-00151-f001:**
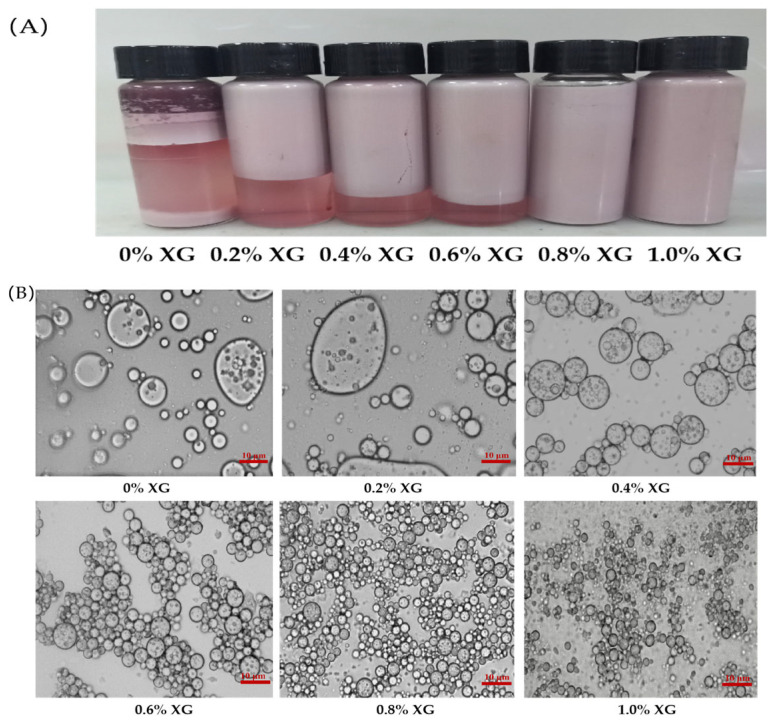
Apparent form (**A**) and optical microstructure under 40× objective magnification (**B**) of double emulsions stabilized by pea protein isolate (PPI)-xanthan gum (XG) mixtures containing 2% (*w*/*v*) of PPI with different concentrations (0%, 0.2%, 0.4%, 0.6%, 0.8% and 1%) of XG.

**Figure 2 foods-12-00151-f002:**
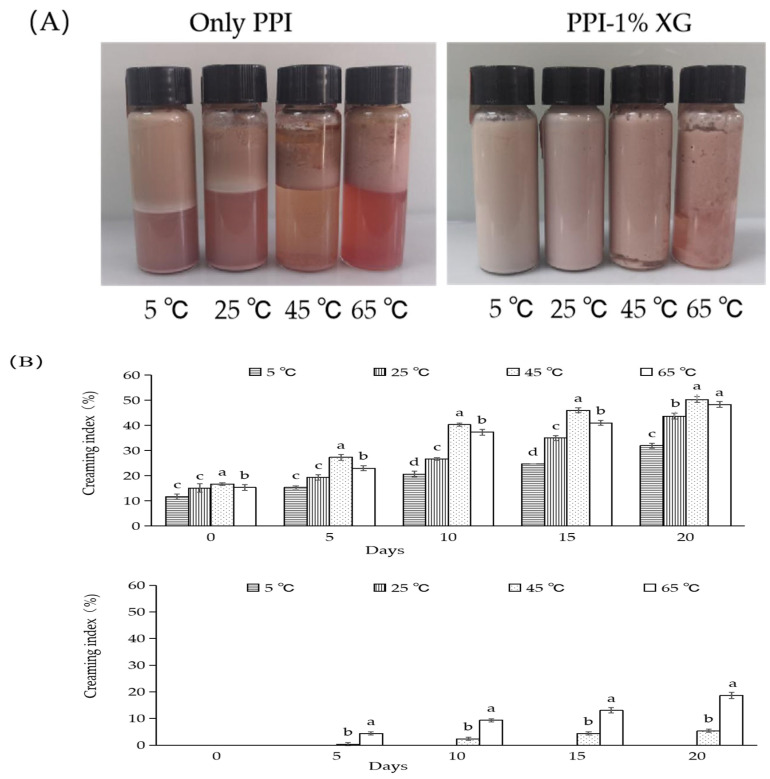
Visual phenomena (**A**), creaming index (**B**) of the double emulsions stabilized by PPI and PPI-1% XG mixtures at different temperatures of 5 °C, 25 °C, 45 °C, 65 °C after 20 days. Different lowercase letters indicated significant differences between double emulsions at different temperatures within a certain storage time (*p* < 0.05).

**Figure 3 foods-12-00151-f003:**
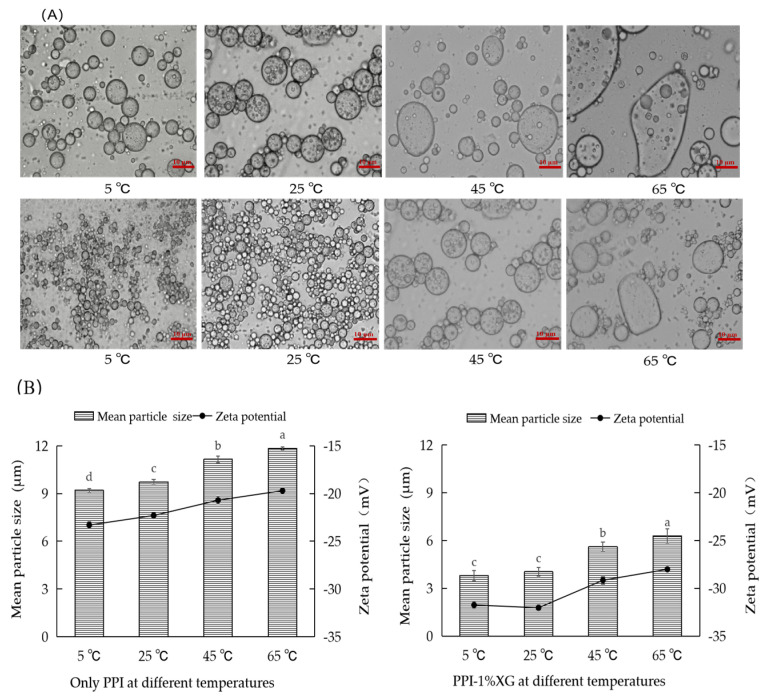
Optical micrograph under 40× objective magnification (**A**), mean particle and zeta potential (**B**) of the double emulsions stabilized by PPI and PPI-1% XG mixture at different temperatures of 5 °C, 25 °C, 45 °C, 65 °C after 20 days. Different lowercase letters indicated significant differences between double emulsions at different temperatures after 20 days (*p* < 0.05).

**Figure 4 foods-12-00151-f004:**
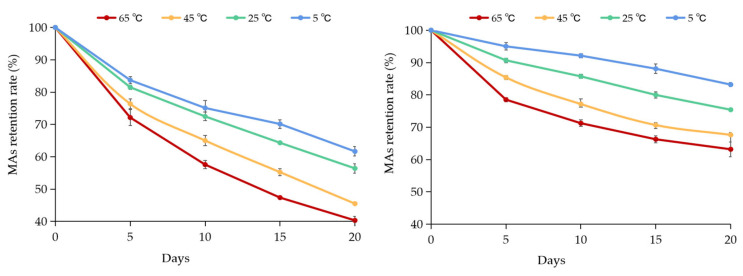
Retention of Mulberry anthocyanins (Mas) of double emulsions stabilized by PPI and PPI-1% XG mixture stored at 5 °C, 25 °C, 45 °C and 65 °C during 20 days.

**Figure 5 foods-12-00151-f005:**
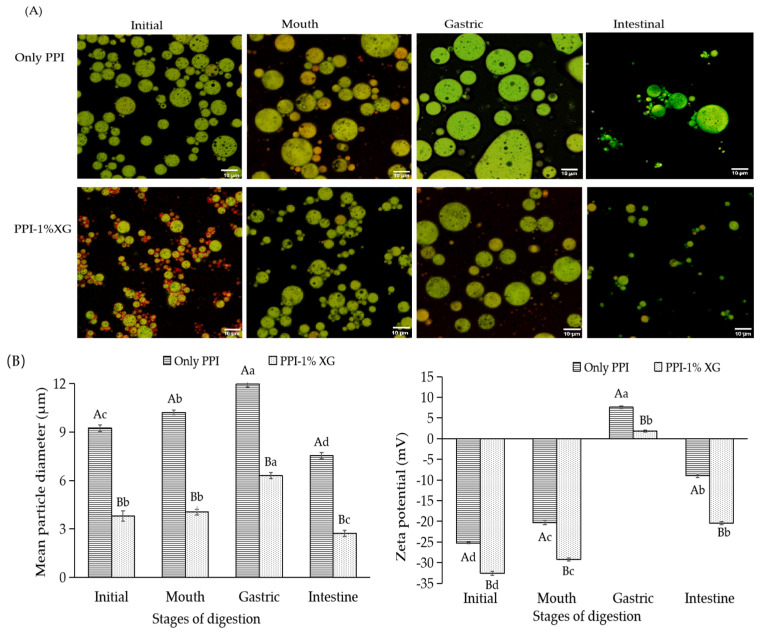
The microscopic fluorescent image under 40× objective magnification (**A**), mean particle and zeta potential (**B**) of double emulsions consisted by PPI and PPI-1% XG during the different stages of in vitro simulated digestion. Different capital letters indicated significant differences (*p* < 0.05) between double emulsions composed of PPI Only and PPI-1% XG at the same in vitro digestion stage. Different lowercase letters indicated significant differences at different in vitro digestion stages of the same double emulsion (*p* < 0.05).

**Figure 6 foods-12-00151-f006:**
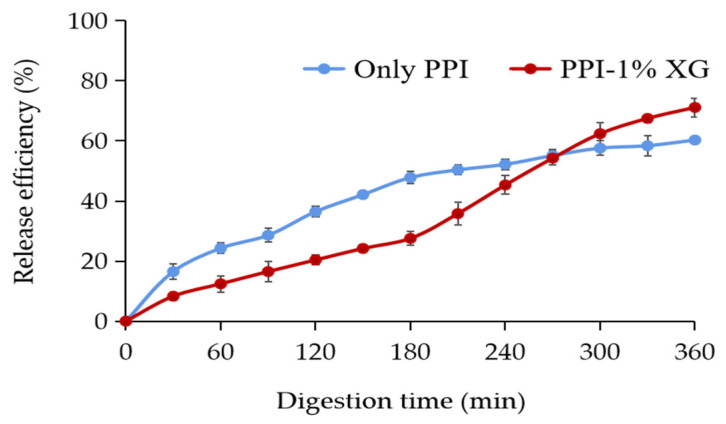
Release efficiency of MAs from double emulsions composed of PPI and PPI-1% XG at the different stages of in vitro simulated digestion.

**Figure 7 foods-12-00151-f007:**
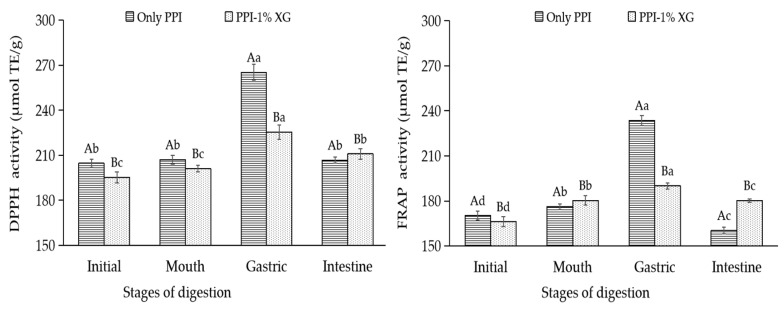
DPPH and FRAP activity of double emulsions composed of PPI or PPI-1% XG at different stages of in vitro simulated digestion. Different capital letters indicated significant differences (*p* < 0.05) between double emulsions composed of PPI Only and PPI-1% XG at the same in vitro digestion stage. Different lowercase letters indicated significant differences at different in vitro digestion stages of the same double emulsion (*p* < 0.05).

**Table 1 foods-12-00151-t001:** Physical characteristics of double emulsions containing 2% (*w*/*v*) of pea protein isolate (PPI) with different concentrations of xanthan gum (XG). The different lowercase letters indicated the significant difference (*p* < 0.05) between the double emulsion with different concentrations of XG added.

	0% XG	0.2% XG	0.4% XG	0.6% XG	0.8% XG	1% XG
Mean particle size (μm)	9.20 ± 0.21 ^b^	9.83 ± 0.17 ^b^	7.40 ± 0.23 ^c^	5.70 ± 0.31 ^d^	4.20 ± 0.24 ^e^	3.80 ± 0.33 ^e^
Zeta potential (mV)	−25.23 ± 0.21 ^a^	−27.60 ± 0.36 ^b^	−28.50 ± 0.30 ^c^	−29.83 ± 0.15 ^d^	−31.73 ± 0.31 ^e^	−32.58 ± 0.31 ^f^
Encapsulation efficiency (%)	44.57 ± 0.93 ^f^	51.50 ± 0.87 ^e^	62.66 ± 0.79 ^d^	68.52 ± 0.81 ^c^	83.02 ± 0.70 ^b^	88.52 ± 2.05 ^a^
Creaming index (%)	74.33 ± 1.53 ^a^	45.00 ± 1.00 ^b^	22.67 ± 1.53 ^c^	12.33 ± 1.15 ^d^	0	0

**Table 2 foods-12-00151-t002:** Degradation regularity of mulberry anthocyanins (MAs) from double emulsions with different temperatures of 5 °C, 25 °C, 45 °C, 65 °C during 20 days. Different lowercase letters indicated significant differences between double emulsions at different temperatures during 20 days (*p* < 0.05).

Temperature	k × 10^−2^ (day^−1^)		t_1/2_ (day)		R^2^	
	PPI	PPI-1% XG	PPI	PPI-1% XG	PPI	PPI-1% XG
5 °C	2.29 ± 0.05 ^d^	0.89 ± 0.05 ^d^	30.32 ± 0.69 ^a^	78.07 ± 4.72 ^a^	0.9716	0.9991
25 °C	2.75 ± 0.06 ^c^	1.18 ± 0.03 ^c^	25.21 ± 0.56 ^c^	60.25 ± 1.33 ^b^	0.9804	0.9936
45 °C	3.80 ± 0.07 ^b^	1.94 ± 0.10 ^b^	18.25 ± 0.34 ^d^	35.73 ± 1.86 ^c^	0.9826	0.9685
65 °C	4.48 ± 0.04 ^a^	2.37 ± 0.20 ^a^	15.40 ± 0.06 ^b^	29.30 ± 2.05 ^d^	0.9803	0.9949

## Data Availability

Data are contained in the article.

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
