# Peer review of "Improved Stabilization and In Vitro Digestibility of Mulberry Anthocyanins by Double Emulsion with Pea Protein Isolate and Xanthan Gum"

_foods, 2022, doi:10.3390/foods12010151_

Round 1

Reviewer 1 Report

Improved Stabilization and In Vitro Digestibility of Mulberry Anthocyanins by Double Emulsion with Pea Protein Isolate and Xanthan Gum

General Issues

Lines 235: “Figure 1” representing optical microscopy image can be mentioned as Figure 1A (which is more specific) and follow the same format at all possible place of figure representations)

Lines 254: Error in the zeta potential of 1%XG addition (does not coincide with table value)

Line 256: the isoelectric point (PI) of PPI (PPI of 4.6)? use PPI and PI correctly for a better understanding

Lines 244 to 246: Need to paraphrase for better understanding

Line 274 and 275 Overlapping of words

Line 277: Under 3.1.4, why was the emulsion index mentioned under the creaming index (are the emulsion index and creaming index the same??)

Line 281:  unit of creaming index (mention its unit wherever required)

Line 284:  How much of XG concentration (%) added improved electrostatic repulsion and more information on the kind of protein, emulsion obtained can be included

Line 311: Is delamination characteristic of double emulsion? If so, add it to the introduction

Line 316: the different temperatures can be written without brackets for readability. Check at all possible places

Lines 321 and 322: Rephrase for better reading

Line 351& 355: Why hyphen between the following temperature values “5-25 °C,” and 45 °C-65 °C? Is that representing the range between two temperatures? If so, why?

Line 359 to 372: These lines are to be checked for grammar, readability

Line 362 PPI emulsion or PPI only double emulsion?

Line 389: What does storage time mean here?

Line 391 and 392: unit of MAs retention after thermal degradation (how much of MAs retained?)

Line 448: Sentence can be rewritten such that the previous study supports the current study

Line 451: The figure number can be mentioned here

Line 453:  Justify the statement “there was significant aggregation between the emulsion droplets” with figure 5.

Under section 2.2, why different protein isolate concentrations were not taken, like XG concentrations? Or justify why 2% of PPI was preferred?

3.1.3 Encapsulation Efficiency: What is the component present in XG that increases the encapsulation efficiency of MAs? How different concentrations of XG positively correlate with encapsulation efficiency

3.1.2 Zeta potential: Research papers supporting the correlation of zeta potential and emulsion stability are to be cited (as mentioned in sections 3.1.3 and 3.1.4)

Section 3 Result: The values obtained after analysis need not be written within brackets. For example, … (9.20 ± 0.21) μm to (9.83 ± 0.17) μm. The above values can be written as …9.20 ± 0.21 μm to 9.83 ± 0.17μm

Format Issues

Line 69: error isoelectric point representation

Line139: Wording error

Line 189: Insert field function while representing equations

Line 192: Check the equation and lines overlapping

Line 236,347, 465: spacing error

Lines 245, 535: Spelling errors

Lines 267 & 268: Typo error

Lines 82, 284, 315: Grammar error

Line 315: Super script issues

Line 333: Error in representing temperature unit

Line 363,522: Punctuation errors

Lines 395: Grammar error

Line 397: Conjunction error

Line 466, 501: Grammar error

Line 429: Super script issue (day 1)

Line 510, 522: why the percentage symbol was used differently at different places? It can be used without brackets for legibility.

Equations 1, 2 and 3: Why percentage symbol was added near 100?

Equation 5: error in quoting equation

Figure 4:  Alignment issue in X-axis title

Write “zeta potential” in the same manner at all places (zeta-potential or zeta potential)

Table1 and 2: Check for correct data alignment, super script issues in representing temperature, order of line numbers

Check the format of all the references, as it is not uniform.

Author Response

Point 1: Lines 235: “Figure 1” representing optical microscopy image can be mentioned as Figure 1A (which is more specific) and follow the same format at all possible place of figure representations)

Response 1: Thank you vey much for your circumspection. “Fig. 1” representing optical microscopy image was replaced to “Fig. 1B” and has followed the same format at all possible place of figure representations.

Point 2: Lines 254: Error in the zeta potential of 1%XG addition (does not coincide with table value)

Response 2: Thank you very much for pointing this out. The zeta potential of 1%XG addition has been modified in the revised version and coincided with table value.

Point 3: Line 256: the isoelectric point (PI) of PPI (PPI of 4.6)? use PPI and PI correctly for a better understanding

Point 4: Lines 244 to 246: Need to paraphrase for better understanding

Response 4: Thanks for the referee’s kind advice. The relevant content has been modified and please check in the revised version.

Point 5: Line 274 and 275 Overlapping of words

Response 5: Thank you vey much for your circumspection. The content has been modified and please check in the revised version.

Point 6: Line 277: Under 3.1.4, why was the emulsion index mentioned under the creaming index (are the emulsion index and creaming index the same??)

Response 6: Thank you vey much for your circumspection. The representation of “emulsion index” was not correct. “creaming index” replaced “emulsion index”. Please check in the revised version.

Point 7: Line 281: unit of creaming index (mention its unit wherever required)

Response 7: Thank you very much for pointing this out. Unit of creaming index has been added wherever required.

Point 8: Line 284: How much of XG concentration (%) added improved electrostatic repulsion and more information on the kind of protein, emulsion obtained can be included

Response 8: The relevant content has been added at the section 3.1.4 of revised version.

Point 9: Line 311: Is delamination characteristic of double emulsion? If so, add it to the introduction

Response 9: When the double emulsion is not stable, delamination will occur. Creaming index  could reflect the stability of emulsion. The high creaming index means the serious delamination and unstablilty of emulsion. Related content has been added at the section of 3.1.4 and 3.2.1.

Point 10: Line 316: the different temperatures can be written without brackets for readability. Check at all possible places

Response 10: Thanks for the kind advice. The brackets for readability has been deleted.

Point 11: Lines 321 and 322: Rephrase for better reading

Response 11: Thank you vey much for your circumspection. The relevant content has been modified and please check in the revised version.

Point 12: Line 351& 355: Why hyphen between the following temperature values “5-25 °C,” and 45 °C-65 °C? Is that representing the range between two temperatures? If so, why?

Response 12: Thank you vey much for your circumspection. “5-25 °C,” and “45 °C-65 °C” did not represent the range between two temperatures. They means “5°C and 25°C” and “45°C and 65°C”.  

The related content has been modified at the section 3.2.2 of revised version.

Point 13: Line 359 to 372: These lines are to be checked for grammar, readability

Response 13: Thank you very much for pointing this out. The relevant content has been modified and please check in the revised version.

Point 14: Line 362 PPI emulsion or PPI only double emulsion?

Response 14: Thanks for your circumspection. “PPI only double emulsion” replaced “PPI emulsion”.

Point 15: Line 389: What does storage time mean here?

Response 15: The storage time refers to the storage of emulsion for 20 days. The related content has been modified at the section 3.3 of revised version.

Point 16: Line 391 and 392: unit of MAs retention after thermal degradation (how much of MAs retained?)

Response 16: Thank you very much for pointing this out. The unit of MAs retention rate after thermal degradation has been added. MAs retention rate means the MAs retained.

Point 17: Line 448: Sentence can be rewritten such that the previous study supports the current study

Response 17: Thanks for the referee’s kind advice. The sentence has been revised such that the previous study supports the current study.

Point 18: Line 451: The figure number can be mentioned here

Response 18: Thank you very much for pointing this out. The figure number has been added and the similar problems has been solved in the revised version.

Point 19: Line 453:  Justify the statement“there was significant aggregation between the emulsion droplets” with figure 5.

Response 19: Thank you very much for pointing this out. The figure number (Fig. 5A) has been added in the sentence to justify “there was significant aggregation between the emulsion droplets”.

Point 20: Under section 2.2, why different protein isolate concentrations were not taken, like XG concentrations? Or justify why 2% of PPI was preferred?

Response 20: Thanks for your circumspection. The concentration of polysaccharide such as xanthan gum has a great influence on the stability of the system. Within a certain range, the greater the concentration of xanthan gum, the greater the viscosity and electrostatic interaction with protein. So in this study, we fixed the concentration of protein to investigate the effect of xanthan gum on the stability of the protein-polysaccharide emulsion.

Point 21: 3.1.3 Encapsulation Efficiency: What is the component present in XG that increases the encapsulation efficiency of MAs? How different concentrations of XG positively correlate with encapsulation efficiency

Response 21: Thanks for your circumspection. The relevant content has been added at the section 3.1.3 of revised version.

Point 22: 3.1.2 Zeta potential: Research papers supporting the correlation of zeta potential and emulsion stability are to be cited (as mentioned in sections 3.1.3 and 3.1.4)

Response 22: Thanks for the referee’s kind advice. Research papers supporting the correlation of zeta potential and emulsion stability has been added in the revised version.

Point 23: Section 3 Result: The values obtained after analysis need not be written within brackets. For example, … (9.20 ± 0.21) μm to (9.83 ± 0.17) μm. The above values can be written as …9.20 ± 0.21 μm to 9.83 ± 0.17μm

Response 23: Thank you very much for pointing this out. Brackets has been deleted and the representation of values has been modified in the revised version.

Point 24: Line 69: error isoelectric point representation

Response 24: Thank you very much for pointing this out. Representation of isoelectric point has been revised.

Point 25: Line139: Wording error

Response 25: Thank you very much for pointing this out. The wording error has been revised.

Point 26: Line 189: Insert field function while representing equations

Response 26: Thanks for your circumspection. The representation of equations has been solved, please see the revised version.

Point 27: Line 192: Check the equation and lines overlapping

Response 27: Thanks for your circumspection. The problem of overlapping has been solved.

Point 28: Line 236,347, 465: spacing error

Response 28: Thank you very much for pointing this out. The spacing error has been revised.

Point 29: Lines 245, 535: Spelling errors

Response 29: Thank you very much for pointing this out. Spelling errors has been solved.

Point 30: Lines 267 & 268: Typo error

Response 30: Thank you very much for pointing this out. Typo error has been solved.

Point 31: Lines 82, 284, 315: Grammar error

Response 31: Thank you very much for pointing this out. The grammar error has been solved.

Point 32: Line 315: Super script issues

Response 32: Thank you very much for pointing this out. Super script issues has been solved.

Point 33: Line 333: Error in representing temperature unit

Response 33: Thanks for your circumspection. Representation of temperature unit has been revised.

Point 34: Line 363,522: Punctuation errors

Response 34: Thank you very much for pointing this out. Punctuation errors has been solved.

Point 35: Lines 395: Grammar error

Response 35: Thank you very much for pointing this out. The grammar error has been solved.

Point 36: Line 397: Conjunction error

Response 36: Thank you very much for pointing this out. Conjunction error has been solved in the revised version.

Point 37: Line 466, 501: Grammar error

Response 37: Thank you very much for pointing this out. The grammar error has been solved.

Point 38: Line 429: Super script issue (day 1)

Response 38: Thank you very much for pointing this out. Super script issue has been solved.

Point 39: Line 510, 522: why the percentage symbol was used differently at different places? It can be used without brackets for legibility.

Response 39: Thanks for your circumspection. The percentage symbol has been used in the same way.

Point 40: Equations 1, 2 and 3: Why percentage symbol was added near 100?

Response 40: Thanks for your circumspection. There is no need to add the percentage symbol.The percentage symbol has been deleted.

Point 41: Equation 5: error in quoting equation

Response 41: Thank you very much for pointing this out. The correct equation has been added.

Point 42: Figure 4:  Alignment issue in X-axis title

Response 42: Alignment issue in X-axis title has been solved.

Point 43: Write “zeta potential” in the same manner at all places (zeta-potential or zeta potential)

Response 43: Thanks for your circumspection. “zeta potential” has been written in the same manner at all places.

Point 44: Table1 and 2: Check for correct data alignment, super script issues in representing temperature, order of line numbers

Response 44: Thank you very much for pointing this out. Data alignment issues, super script issues in representing temperature, order of line numbers have been revised.

Point 45: Check the format of all the references, as it is not uniform.

Response 45: Thanks for your circumspection. The format of all the references has been set in the same way.

Reviewer 2 Report

The reviewer believes that this is a well-structured manuscript presenting an interesting set of experimental data on the Stabilization and In Vitro Digestibility of Mulberry Anthocyanins by Double Emulsion with Pea Protein Isolate and Xanthan Gum. The authors provide a comprehensive analysis and discussion of their findings with reference on most available literature reference. Overall, this publication will create added value on this scientific field.

A few comments for the consideration of the authors are given below:

-        Line 66-69: Something wrong on the structure of the sentence?…can you please explain better how the shift to isoelectric point affects microstructural stability? For instance for protein stabilised emulsions, someone may expect that “When the pH approaches the isoelectric point of the protein, the electrostatic repulsion of the protein adsorption layers decreases, which allows more compact packing and stronger attractive interfacial interactions so that coalescence and flocculation occur..”

-        Line 93:  “are” to study… (instead of “is”)?

-        Line 153: Can you please give a literature reference for the DLS method uses for droplet size measurements (unless covered by reference 21/not clear…)  

-        Lines 221-223: I suggest you include a separate section (e.g. 2.10?) as “statistical analysis” where to insert these sentences and also develop a bit. For instance how was the comparison of methods derived (ANOVA test?) and how you present in graphs the reported statistical differences (e.g. order of activity, a>b>c etc?)

-        Line 230: you may elaborate a bit by also covering the oxidative stability for protein stabilised emulsions? For instance, smaller droplet size associate to physically more stable emulsion but also more stable to oxidative destabilisation….In general, smaller oil droplets, obtained in protein emulsions through the increase of protein concentration with no change in processing conditions, are associated with enhanced protection from oxidative changes.

-        Figures: Please ensure that legends for all figures are self-explanatory in terms of results presentation (e.g. results in means + SD, order of activity a>b>c?...significant level, p=?)

-        Lines 253-254: I am not sure whether the term “antioxidant capacity of emulsion” is the most appropriate….do we actually refer here to “enhanced physical or oxidative stability of emulsions”??

-        Conclusions: I would suggest that authors in this session indicate more explicitly for the reader (e.g. in bullet points)

(i)              what is the most novel observations of this manuscript; any need for future work? (e.g. for optimisation of the emulsion system?)

(ii)            is there any potential for product development based on these findings? E.g. a product that will concern an anthocyanins delivery emulsion system? (e.g. market applications for food/pharmaceutical sectors?)…

-        References:

(i)              Please ensure that MDPI guidelines are followed for all listed References (e.g. Nr 7 need correction, bold for year/italics for volume)…also ensure consistency….why journal abbreviation in capital letters?

(ii)            You may wish to also cross check/consider the following review paper since you may find similar areas of investigation and parameters of relevance? (e.g. effect of pH/droplet size on emulsion stability)/

Author Response

Point 1: Line 66-69: Something wrong on the structure of the sentence?…can you please explain better how the shift to isoelectric point affects microstructural stability? For instance for protein stabilised emulsions, someone may expect that “When the pH approaches the isoelectric point of the protein, the electrostatic repulsion of the protein adsorption layers decreases, which allows more compact packing and stronger attractive interfacial interactions so that coalescence and flocculation occur..”

Response 1: Thank you very much for the referee’s kind advice. The relevant content has been revised. Please see line 70-73 in the revised version.

Point 2:    Line 93:  …“are” to study… (instead of “is”)?

Response 2: Thanks for your circumspection. “are” has replaced  “is”.

Point 3: Line 153: Can you please give a literature reference for the DLS method uses for droplet size measurements (unless covered by reference 21/not clear…)  

Response 3: Thank you very much for the referee’s kind advice. Literature reference for the DLS method uses has been added.

Point 4: Lines 221-223: I suggest you include a separate section (e.g. 2.10?) as “statistical analysis”where to insert these sentences and also develop a bit. For instance how was the comparison of methods derived (ANOVA test?) and how you present in graphs the reported statistical differences (e.g. order of activity, a>b>c etc?)

Response 4: Thank you very much for the referee’s kind advice. The new section of “statistical analysis” has added.

Point 5: Line 230: you may elaborate a bit by also covering the oxidative stability for protein stabilised emulsions? For instance, smaller droplet size associate to physically more stable emulsion but also more stable to oxidative destabilisation….In general, smaller oil droplets, obtained in protein emulsions through the increase of protein concentration with no change in processing conditions, are associated with enhanced protection from oxidative changes.

Response 5: Thank you very much for the referee’s kind advice. In our study, we found the significant effect of digestion environment on oxidative stability of double emulsions and found the similar results which showed at section of 3.4.4. However, we did not found much references about the influence of different concentration protein on the oxidative stability of double emulsions. We will study the  oxidative stability for protein stabilised emulsion at different pH value or protein concentrations in the future.

Point 6: Figures: Please ensure that legends for all figures are self-explanatory in terms of results presentation (e.g. results in means + SD, order of activity a>b>c?...significant level, p=?)

Response 6: Thanks for your circumspection. In the revised version, legends for all figures has been  self-explanatory in terms of results presentation. The relevant explanation has been showed at the section of “Statistical Analysis”.

Point 7: Lines 253-254: I am not sure whether the term “antioxidant capacity of emulsion” is the most appropriate….do we actually refer here to “enhanced physical or oxidative stability of emulsions”??

Response 7: Thank you very much for the referee’s kind advice. In the study, we refer to the  “enhanced oxidative stability of emulsions”. So at the sections of 2.9 and 3.4.4 , we changed the term “antioxidant capacity of emulsion” to  “oxidative stability of double emulsions”.

Point 8:  Conclusions: I would suggest that authors in this session indicate more explicitly for the reader (e.g. in bullet points)

(i)       what is the most novel observations of this manuscript; any need for future work? (e.g. for optimisation of the emulsion system?)

(ii)       is there any potential for product development based on these findings? E.g. a product that will concern an anthocyanins delivery emulsion system? (e.g. market applications for food/pharmaceutical sectors?)…

Response 8: 

  • Thank you very much for the referee’s kind advice. The content about the novel observations and future work has been added in the revised conclusions.   
  • Thank you very much for the referee’s kind advice. The content about the potential for product development based on these findings has been added in the revised conclusions.

Point 9:  References:

(i)              Please ensure that MDPI guidelines are followed for all listed References (e.g. Nr 7 need correction, bold for year/italics for volume)…also ensure consistency….why journal abbreviation in capital letters?

(ii)            You may wish to also cross check/consider the following review paper since you may find similar areas of investigation and parameters of relevance? (e.g. effect of pH/droplet size on emulsion stability)/

 Response 9: 

  • Thank you very much for The format of all the references has been set consistency. Please see the revised version. According to the MDPI guidelines, the wrong representation of journal name has been modified.

  • Thank you very much for the referee’s kind advice. There are some similar areas of investigation about the pH and ionic strength on emulsion stability. These relevant parameters will be considered later.
